# Subsequent Upper Urinary Tract Carcinoma Related to Worse Survival in Patients Treated with BCG

**DOI:** 10.3390/cancers15072002

**Published:** 2023-03-28

**Authors:** Kazuyuki Numakura, Makito Miyake, Mizuki Kobayashi, Yumina Muto, Yuya Sekine, Nobutaka Nishimura, Kota Iida, Masanori Shiga, Shuichi Morizane, Takahiro Yoneyama, Yoshiaki Matsumura, Takashige Abe, Takeshi Yamada, Kazumasa Matsumoto, Junichi Inokuchi, Naotaka Nishiyama, Rikiya Taoka, Takashi Kobayashi, Takahiro Kojima, Hiroshi Kitamura, Hiroyuki Nishiyama, Kiyohide Fujimoto, Tomonori Habuchi

**Affiliations:** 1Department of Urology, Akita University Graduate School of Medicine, Akita 010-8543, Japan; 2Department of Urology, Nara Medical University, Kashihara 634-8521, Japan; 3Department of Urology, Faculty of Medicine, University of Tsukuba, Tsukuba 305-8576, Japan; 4Division of Urology, Faculty of Medicine, Tottori University, Tottori 683-8504, Japan; 5Department of Urology, Hirosaki University Graduate School of Medicine, Hirosaki 036-8563, Japan; 6Department of Urology, Nara Prefecture General Medical Center, Nara 630-8054, Japan; 7Department of Renal and Genitourinary Surgery, Graduate School of Medicine, Hokkaido University, Sapporo 060-8648, Japan; 8Department of Urology, Kyoto Prefectural University of Medicine, Kyoto 602-8566, Japan; 9Department of Urology, Kitasato University School of Medicine, Sagamihara 252-0373, Japan; 10Department of Urology, Graduate School of Medical Sciences, Kyushu University, Fukuoka 812-8582, Japan; 11Department of Urology, Faculty of Medicine, University of Toyama, Toyama 930-0194, Japan; 12Departments of Urology, Kagawa University Faculty of Medicine, Takamatsu 761-0793, Japan; 13Department of Urology, Kyoto University Graduate School of Medicine, Kyoto 606-8507, Japan; 14Department of Urology, Aichi Cancer Center, Nagoya 464-8681, Japan

**Keywords:** BCG, non-muscle-invasive bladder cancer, upper urinary tract cancer, intravesical recurrence, multiple bladder tumors

## Abstract

**Simple Summary:**

This retrospective cohort study aimed to understand the incidence, clinical impact, and risk factors associated with upper urinary tract urothelial carcinoma (UTUC) after intravesical Bacillus Calmette-Guerin (BCG) therapy. The study included 3226 patients diagnosed with non-muscle-invasive bladder cancer (NMIBC) and treated with intravesical BCG therapy between January 2000 and December 2019. Of these patients, 6.1% were diagnosed with UTUC during the follow-up period, and those with UTUC had worse survival rates compared to those without UTUC. Tumor multiplicity, treatment for Connaught strain, and intravesical recurrence after BCG therapy were associated with subsequent UTUC diagnosis. The study suggests that patients with these risk factors may require closer monitoring for UTUC after BCG therapy.

**Abstract:**

Upper urinary tract urothelial carcinoma (UTUC) after intravesical bacillus Calmette-Guerin (BCG) therapy is rare, and its incidence, clinical impact, and risk factors are not fully understood. To elucidate the clinical implications of UTUC after intravesical BCG therapy, this retrospective cohort study used data collected between January 2000 and December 2019. A total of 3226 patients diagnosed with non-muscle-invasive bladder cancer (NMIBC) and treated with intravesical BCG therapy were enrolled (JUOG-UC 1901). UTUC impact was evaluated by comparing intravesical recurrence-free survival (RFS), cancer-specific survival (CSS), and overall survival (OS) rates. The predictors of UTUC after BCG treatment were assessed. Of these patients, 2873 with a medical history that checked UTUC were analyzed. UTUC was detected in 175 patients (6.1%) during the follow-up period. Patients with UTUC had worse survival rates than those without UTUC. Multivariate analyses revealed that tumor multiplicity (odds ratio [OR], 1.681; 95% confidence interval [CI], 1.005–2.812; *p* = 0.048), Connaught strain (OR, 2.211; 95% CI, 1.380–3.543; *p* = 0.001), and intravesical recurrence (OR, 5.097; 95% CI, 3.225–8.056; *p* < 0.001) were associated with UTUC after BCG therapy. In conclusion, patients with subsequent UTUC had worse RFS, CSS, and OS than those without UTUC. Multiple bladder tumors, treatment for Connaught strain, and intravesical recurrence after BCG therapy may be predictive factors for subsequent UTUC diagnosis.

## 1. Introduction

The urinary bladder is the predominant organ affected in primary urothelial carcinoma (UC), although UC can arise throughout the urinary tract. Upper urinary tract urothelial carcinoma (UTUC) accounts for 5% to 10% of all urothelial carcinomas [1]. Although UTUC and UC in the bladder (UCB) have previously been considered to share histopathological features, remarkable clinical and molecular differences exist between cancer sites [2]. Approximately 60% of patients with UTUC have an invasive tumor versus 20% to 25% of patients with UCB [3]. The prognosis of UTUC is poor, with a 5-year overall survival (OS) of approximately 70%. For invasive disease, the 5-year OS is less than 40%, which is worse than the 5-year OS in patients with muscle-invasive UBC treated with radical cystectomy [4,5].

After intravesical bacillus Calmette-Guerin (BCG) instillation therapy, UTUC is diagnosed in 4%–9% of patients [3,6]. This rate was higher than that of UTUC among all patients with UBC [7] and was equivalent to the recurrence rate after total cystectomy [8,9]. Clinical risk factors for the development of subsequent UTUC are tumor multiplicity, tumor location (involved in the ureteral orifice), advanced tumor stage, the existence of carcinoma in situ (CIS), and operative modality [10]. Although every guideline (the National Comprehensive Cancer Network, the American Urological Association, and the European Association of Urology) recommends careful follow-up of the entire urinary tract after treatment for high-grade UC, diagnosis of subsequent UTUC has often been delayed [11]. Indeed, little is known about the clinical implications of subsequent UTUC after BCG instillation for UBC.

In this study, we attempted to elucidate the clinical features of subsequent UTUC after BCG therapy and to elucidate the risk factors for its development in a retrospective cohort of 3226 patients in Japan (JUOG-UC 1901).

## 2. Materials and Methods

### 2.1. Data Collection and Study Cohort

This retrospective multicenter study was approved by the institutional review board of each participating institute (reference protocol ID: 2266 in the IRB of Akita University) of the Japan Urological Oncology Group framework. Informed consent was obtained from the participants through posters and/or websites following the opt-out policy (https://www.lifescience.mext.go.jp/bioethics/seimeikagaku_igaku.html) (last access date 19 March 2023). We recruited 3226 patients who received intravesical BCG therapy for NMIBC treated between 2000 and 2019 at 31 hospitals in Japan. The clinical characteristics of the patients were investigated, including age; sex; performance status; former history of NMIBC; tumor multiplicity; tumor size; T category; tumor grade (per the 2004 World Health Organization classification); second transurethral resection (TUR); presence of bladder CIS and prostatic urethra-involving CIS; divergent differentiation such as squamous differentiation and glandular differentiation; variant histology such as nested, micropapillary, and plasmacytoid variants; and lymphovascular involvement (LVI).

Of the 3226 patients, 353 (11%) were excluded due to missing data on subsequent UTUC status; thus, 2873 (89%) were analyzed. Appendix A shows the flowchart of the patient selection process.

### 2.2. Intravesical BCG Treatment after Transurethral Resection of Bladder Tumor (TURBT)

The criteria, dose, and schedule for initial BCG and maintenance BCG therapy were not strictly scheduled and were given as clinical implementation by each physician’s decision. Most eligible patients were at high or highest risk of NMIBC, such as those who had papillary Ta/T1 high-grade tumors and CIS, and were treated with intravesical BCG therapy after TURBT. The intravesical BCG therapy consisted of weekly instillations of Immunobladder (80 mg of Tokyo-172 strain) or ImmuCyst (81 mg of Connaught strain, currently unavailable) for 6–8 consecutive weeks, with or without subsequent maintenance BCG therapy.

### 2.3. Surveillance

The surveillance protocol varied depending on the policies of the individual institutions and the number of physicians. In general, patients underwent check-ups by white-light cystoscopy and urinary cytology every 3 months for the first 2 years, every 4 months in the third year, every 6 months in the fourth and fifth years, and annually thereafter. This follow-up protocol followed a conventional Japanese style and may be stricter than the established guidelines [12,13,14]. Recurrence was defined as recurrent tumors of pathologically proven urothelial carcinoma in the bladder and/or prostatic urethra. Progression was defined as recurrent disease with invasion of the muscularis propria (≥T2), positive regional lymph nodes, and/or distant metastases. UTUC was diagnosed based on a positive finding on a computed tomography scan of the upper urinary tract (UUT), ipsilateral positive urine cytology, and ureteroscopic pathological diagnosis.

### 2.4. Statistical Analysis

Clinicopathological characteristics were compared using Mann–Whitney U, chi-square, and Kruskal–Wallis tests as appropriate. Intravesical recurrence-free survival (RFS), cancer-specific survival (CSS), and OS were calculated from the date of initial induction of BCG treatment. Survival rates were analyzed using the Kaplan–Meier method, and significance was compared using the Cox hazard regression model. To elucidate the risk factors for subsequent UTUC and reduce the effects of selection bias and potential confounders in this observational study, we performed multiple logistic regression analyses. A multivariate analysis was used for the analysis of odds ratios (ORs) and 95% confidence intervals (CIs) for factors with *p* < 0.1 in the univariate analysis. Statistical analyses were performed using the SPSS statistical software (version 26.0; SPSS Japan Inc., Tokyo, Japan). All reported *p* values were two-sided, and statistical significance was set at *p* < 0.05.

## 3. Results

### 3.1. Patient Characteristics

Of the 3226 patients with NMIBC who were treated with BCG therapy, 2873 were analyzed, and 175 (6.1%) were diagnosed with UTUC during the follow-up period from the initiation of BCG therapy (Table 1). In this patient population, the median period of total observation was 48.0 (7–215) months, and the median period between the first dose of the BCG treatment and diagnosis with UTUC was 27.8 (3–182) months. There were 132 men and 43 women in the UTUC group; the median age was 71.0 (28–86) years when UTUC was confirmed. The location of urothelial carcinoma was the renal pelvis in 33 patients, the ureter in 106 patients, both the pelvis and the ureter in 8 patients, and not described in 27 patients. The pathological findings of the preceding TURBT were Ta or T1 in 124 patients and Tis in 51 patients (Table 1).

### 3.2. Survival Analyses

Regarding the comparison of survival rates between patients with and without subsequent UTUC, RFS (median 70.5 m vs. 147.3 m, HR 2.552, 95% CI 2.069–3.149, *p* < 0.001), CSS (median 168.3 m vs. 227.4 m, HR 3.434, 95% CI 2.284–5.162, *p* < 0.001), and OS (median 156.0 m vs. 189.6 m, HR 1.485, 95% CI 1.063–2.075, *p* = 0.020) were worse for patients with UTUC (Figure 1).

### 3.3. Analyses of Predictive Factors for Subsequent UTUC after BCG

We assessed the clinical factors to predict UTUC after BCG therapy and performed a univariate analysis (Table 2). Bladder recurrence after BCG therapy (OR 4.200, 95% CI 3.057–5.770, *p* < 0.001), high-grade carcinoma (OR 2.489, 95% CI 1.211–5.116, *p* = 0.009), multiple bladder tumors (OR 1.980, 95% CI 1.313–2.986, *p* = 0.001), bladder tumor >3 cm (OR 1.815, 95% CI 1.182–2.787, *p* = 0.009), Connaught strain (OR 1.789, 95% CI 1.290–52.183, *p* = 0.001), CIS (OR 1.758, 95% CI 1.252–2.470, *p* = 0.002), female sex (OR 1.577, 95% CI 1.102–2.256, *p* = 0.018), and smoking history (OR 1.504, 95% CI 1.085–2.088, *p* = 0.017) were identified as risk factors for subsequent UTUC (Table 2).

To reduce confounding factors, we applied multiple logistic regression analyses to these risk factors from the univariate analysis. The independent factors for subsequent UTUC after BCG therapy were intravesical recurrence (OR 5.097, 95% CI, 3.225–8.056; *p* < 0.001), Connaught strain (OR 2.211, 95% CI, 1.380–3.543; *p* = 0.001), and multiple tumors at TURBT (OR 1.681, 95% CI, 1.005–2.812; *p* = 0.048) (Table 3).

## 4. Discussion

In this retrospective, multi-institutional study, subsequent UTUC was detected in 175 (6.1%) patients during the follow-up period after BCG instillation therapy for NMIBC. Patients with UTUC had significantly poorer RFS, CSS, and OS rates than those without UTUC. Intravesical recurrence after the initial BCG therapy, Connaught strain, and tumor multiplicity were associated with a subsequent UTUC diagnosis.

The rate of a subsequent UTUC diagnosis of 6.1% in our retrospective study was equivalent to previous reports about the UTUC rate after total cystectomy [8,9] and after BCG bladder instillation therapy [3,6]. These rates were higher than those of subsequent UTUC in all UBC populations. To explain this high rate, two possible explanations have been debated, namely, the “field effect” and “tumor seeding” hypotheses, concerning UTUC incidence after UBC [9,15,16]. However, recent next-generation sequencing (NGS) has identified significant clonality between primary UTUC and metachronous UBC in individual patients, strongly suggesting that tumor seeding rather than de novo oncogenesis drives recurrences [17]. This result raises the argument that clinicians might fail to detect UTUC at the time of UBC diagnosis [11]. Further evaluation, such as a paired analysis using NGS between specimens of UBC and subsequent UTUC, might resolve this fundamental question.

Subsequent UTUC had a negative impact on survival rates in our analysis. Although the association between prognosis and UTUC after BCG treatment has been controversial, our study recruited a significant number of patients compared to previous studies [6,11,18,19,20]. In general, UTUC after BCG therapy might have a clinical risk of a poor survival rate, together with higher tumor stage [20], LVI, and Tis [11] in patients with UBC before BCG treatment. However, our study showed that these risk factors were not associated with a subsequent UTUC diagnosis. Patients with UTUC after cystectomy showed poorer survival rates, which might have directly contributed to poor prognoses in our study.

Connaught strain was identified as a risk factor for subsequent UTUC in our study; however, we do not believe this is a general risk factor worldwide. In terms of bladder recurrence rate, no obvious difference in treatment efficacy among strains has been reported. However, in the patients who were treated with two courses or more [21], general symptoms such as fever were more frequently seen in the patients treated with the Tokyo strain as the first course and the second course compared to patients treated with the Tokyo strain as the first course and the Connaught strain as the second course. This result may help us speculate that the treatment sequence of the same strain affects distant lesions more effectively. The acquired immunology could be boosted by BCG instillation in the Japanese strain because quite a few Japanese were vaccinated with the BCG Tokyo strain during their school days to prevent tuberculosis [22]. This effort might increase treatment efficacy for patients with urothelial carcinoma of the whole urinary tract when they receive BCG bladder instillation therapy.

Our results showed a statistically significant difference in subsequent UTUC in patients with multiple diseases compared to those with a solitary lesion at TURBT before BCG bladder instillation. These multiple diseases might have multifocal abnormalities not only in the bladder but also in the whole urinary tract [7]. Tumor recurrence in the bladder is a major risk factor for tumor progression after BCG [23] and is also reported as a risk factor for subsequent UTUC [6]. There could be some explanations for this relationship. First, early recurrence of UBC after BCG therapy, which was recently defined as “BCG unresponsive” is a high-risk entity in local and systemic disease progression. This risky entity could lead to multifocal disease in the whole urinary tract and could be associated with field effect theory [24]. Second, primary UTUC might already exist during BCG therapy and emit cancer cells into the bladder; this is known as tumor seeding [15]. Schwab et al. [18] reported that 75 patients with positive urine cytology in the absence of visible bladder tumors after a complete response to BCG therapy showed 83% disease recurrence within the urinary tract. Of these recurrences, 20% were detected as UTUCs. Low-grade UTUC might exist more often with negative cytology and be invisible even in recent imaging modality [25,26]. Indeed, patients with UTUC following radical cystectomy have poorer outcomes, even with radical nephroureterectomy.

Although recurrence of UC in the remnant urothelium is a rare event (4%–10% in the upper urinary tract), most patients have an adverse prognosis, despite the absence of distant disease at diagnosis [9]. Therefore, surveillance of the remnant urothelium should be carried out for patients, as it may improve early detection and confer therapeutic benefits. In patients treated with cystectomy, subsequent UC developed mainly in those with risk factors for recurrence [27]. As the number of risk factors has been shown to affect the incidence of subsequent UC, the intensity of surveillance should be based on a risk-adapted strategy [27]. This suggests that patients should be intensively examined for recurrence after BCG therapy, being treated with a non-standard BCG strain, and multiple tumors at TURBT [9]. In patients with risk factors, surveillance, such as urine analysis, urine cytology, and ultrasound sonography, should be conducted at least annually for pan-urothelial disease years after BCG. Diagnostic urethroscopy and cross-sectional imaging of the upper tract should be performed in cases of suspected positivity in screenings [9].

The present study had several limitations. First, since this was a retrospective chart review study, some important information was lacking, such as the reason UTUC was diagnosed, UTUC development after BCG therapy, BCG dose modification details, treatment discontinuation, and concomitant drug therapy. Second, the BCG strain had to be switched from the Connaught strain to the Tokyo strain in the middle of the study because of the supply limitation of the Connaught strain. This change might have affected patient outcomes. Third, the superiority of BCG maintenance therapy since its conception makes it a standard practice today, as it results in less recurrence and improved progression rates in patients with NMIBC with intermediate-to-high risk. However, only 17.9% of the patients in our cohort received it. Finally, there may be a regional bias, and our results may not be generalizable to other populations due to differences in medical practices. Nevertheless, to the best of our knowledge, this is the largest study to report UTUC after BCG instillation therapy in patients with NMIBC.

## 5. Conclusions

Patients with NMIBC with subsequent UTUC showed worse survival outcomes than those without UTUC after BCG bladder instillation therapy. Intravesical recurrence after BCG treatment by Connaught strain and multiple bladder tumors were risk factors for subsequent development of UTUC.

## Figures and Tables

**Figure 1 cancers-15-02002-f001:**
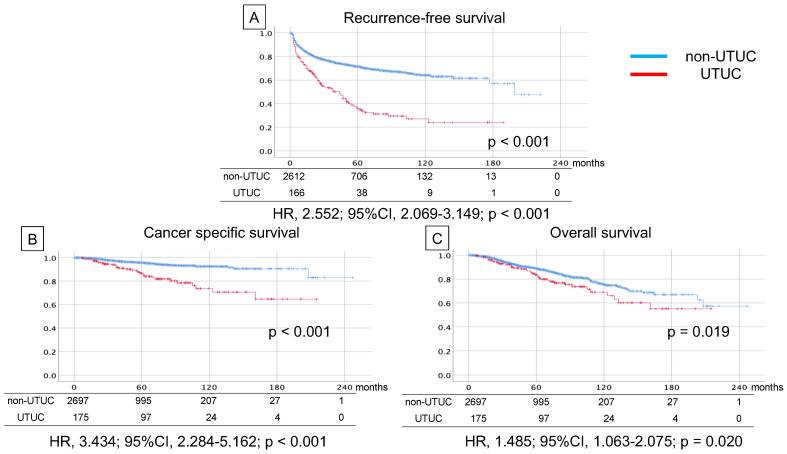
Kaplan–Meier curve of recurrence-free survival (**A**), cancer-specific survival (**B**), and overall survival (**C**) in non-muscle-invasive bladder cancer patients with and without subsequent diagnosis of upper tract urothelial carcinoma after intravesical bacillus Calmette-Guerin (BCG) therapy.

**Table 1 cancers-15-02002-t001:** Characteristics of patients with subsequent diagnosis with UTUC after BCG bladder instillation therapy.

	Subsequent Diagnosis with UTUC (N = 175, 5.4%)	
Observation duration	Months (range)	27.8 (0–182)
Gender	Male	132
	Female	43
Age	Median (range)	71.0 (28.0–86)
Location of UTUC	Renal pelvis	33
	Ureter	105
	Renal pelvis + ureter	8
	Not described	27
Primary bladder T stage	a or 1	124
	CIS	51

UTUC, Urinary tract urothelial carcinoma; BCG, bacillus Calmette-Guerin; T, clinical tumor stage; CIS, carcinoma in situ.

**Table 2 cancers-15-02002-t002:** Clinical predictive factors for subsequent UTUC after BCG bladder instillation therapy in univariate analysis.

					95% CI		
		Subsequent UTUC	No UTUC	OR	Lower	Upper	*p*
Gender	Male:Female	132:43	2236:462	1.577	1.102	2.256	0.018
Age	Median (range)	71.0 (28.0–86.0)	72.0 (29.0–97.0)	-	-	-	0.499
PS	0:1 or More	151:15	2317:278	0.828	0.480	1.428	0.603
Smoking	Previous and present:No	77:76	1461:959	1.504	1.085	2.088	0.017
Grade	High:Low	159:8	2324:291	2.489	1.211	5.116	0.009
T	0 and 1:CIS	124:51	2189:512	1.758	1.252	2.470	0.002
Concurrent CIS	Yes:No	81:37	1030:591	1.256	0.840	1.877	0.277
Multiplicity	Multiple:Solitary	130:29	1825:806	1.980	1.313	2.986	0.001
Maximum diameter (cm)	3 or more:Less than 3	33:68	438:1638	1.815	1.182	2.787	0.009
Appearance	Papillary:Others	46:114	648:1973	1.213	-	-	0.259
Variant histology	UC:Others	5:170	63:2619	1.206	-	-	0.607
Lympho-vascular invasion	Yes:No	10:164	117:2581	1.345	0.692	2.615	0.343
BCG strain	Connaught:Tokyo	59:114	603:2086	1.789	1.290	2.183	0.001
Incomplete BCG induction therapy	Yes:No	19:156	348:2353	0.824	-	-	0.485
Maintenance Therapy	Yes:No	24:151	493:2208	0.712	0.458	1.107	0.154
Recurrence in bladder	Yes:No	108:66	752:1930	4.200	3.057	5.770	<0.001

UTUC, Upper urinary tract carcinoma; BCG, bacillus Calmette-Guerin; PS, performance status; T, clinical tumor stage; CIS, carcinoma in situ; UC, urothelial carcinoma; OR, odds ratio; CI, confidence interval.

**Table 3 cancers-15-02002-t003:** Multivariable analysis for elucidating a risk factor of subsequent UTUC by logistic regression analysis.

Factor	Risk Category	Multivariable			
		OR	95% CI		*p*
			Lower Limit	Upper Limit	
Gender	female	1.644	0.938	2.881	0.082
Smoking history	yes	1.490	0.917	2.427	0.108
Grade	high	1.405	0.591	3.342	0.442
T	CIS	1.217	0.633	2.339	0.557
Multiplicity	multiple	1.681	1.005	2.812	0.048
Tumor diameter	3 cm≤	1.610	1.001	2.591	0.055
Strain	Connaught	2.211	1.380	3.543	0.001
Intravesical recurrence	yes	5.097	3.225	8.056	<0.001

UTUC, Upper urinary tract carcinoma; T, clinical tumor stage; CIS, carcinoma in situ; OR, odds ratio; CI, confidence interval.

## Data Availability

The data presented in this study are available on request from the corresponding author.

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
