# Peer review of "Subsequent Upper Urinary Tract Carcinoma Related to Worse Survival in Patients Treated with BCG"

_cancers, 2023, doi:10.3390/cancers15072002_

Round 1

Reviewer 1 Report

In the simple summary the authors have said - European population, where as this is Japanese population. ' In this report we looked at the outcomes from invasive bladder cancer in a large European population'. Please amend.

In the discussion, some mention should be done on diagnostic aspects of UTUC and endoscopic management of UTUC. In this regard they should also quote this reference:

1. Villa L, Haddad M, et al. Which Patients with Upper Tract Urothelial Carcinoma Can be Safely Treated with Flexible Ureteroscopy with Holmium:YAG Laser Photoablation? Long-Term Results from a High Volume Institution. DOI: 10.1016/j.juro.2017.07.088

Author Response

In the simple summary the authors have said - European population, where as this is Japanese population. ' In this report we looked at the outcomes from invasive bladder cancer in a large European population'. Please amend.

Response: We are sorry that we did a terrible mistake. We totally rewrote the simple summary as follows:

Survivals of subsequent UTUC from NMIBC treated by BCG were worse. Multiple bladder tumors, Connaught strain, and intravesical recurrence may predict subsequent UTUC.

In the discussion, some mention should be done on diagnostic aspects of UTUC and endoscopic management of UTUC. In this regard they should also quote this reference:

  1. Villa L, Haddad M, et al. Which Patients with Upper Tract Urothelial Carcinoma Can be Safely Treated with Flexible Ureteroscopy with Holmium:YAG Laser Photoablation? Long-Term Results from a High Volume Institution. DOI: 10.1016/j.juro.2017.07.088

Response: Thank you for your kind suggestion. We referred to the paper from line 220 as reference number 26.

Reviewer 2 Report

This manuscript clearly presents data from a well designed and executed study on a very interesting topic. It is accepted for publication.

Author Response

We appreciate reviewer #2. We feel all of your contributions have helped us significantly improve the paper. We hope we will work with you again in the future.